# Faster Minimum Bayes Risk Decoding with Confidence-based Pruning

**Julius Cheng, Andreas Vlachos**
Department of Computer Science and Technology
University of Cambridge
{jncc3,av308}@cam.ac.uk

## Abstract

Minimum Bayes risk (MBR) decoding outputs the hypothesis with the highest expected utility over the model distribution for some utility function. It has been shown to improve accuracy over beam search in conditional language generation problems and especially neural machine translation, in both human and automatic evaluations. However, the standard sampling-based algorithm for MBR is substantially more computationally expensive than beam search, requiring a large number of samples as well as a quadratic number of calls to the utility function, limiting its applicability. We describe an algorithm for MBR which gradually grows the number of samples used to estimate the utility while pruning hypotheses that are unlikely to have the highest utility according to confidence estimates obtained with bootstrap sampling. Our method requires fewer samples and drastically reduces the number of calls to the utility function compared to standard MBR while being statistically indistinguishable in terms of accuracy. We demonstrate the effectiveness of our approach in experiments on three language pairs, using chrF++ and COMET as utility/evaluation metrics.

## 1 Introduction

Minimum Bayes risk (MBR) decoding (Bickel and Doksum, 1977; Goel and Byrne, 2000) has recently gained renewed attention as a decision rule for conditional sequence generation tasks, especially neural machine translation (NMT). In MBR, the sequence with the highest expected utility with respect to thez model distribution is chosen as the output, where the utility is usually some measure of text similarity. This contrasts with the more commonly used maximum a posteriori (MAP) decision rule, which returns the sequence with the highest probability under the model. MAP is generally intractable, and beam search is typically used to find an approximation. MBR is likewise intractable,

and Eikema and Aziz (2020) propose an sampling-based approximation algorithm.

MBR has been shown to outperform MAP beam search in both automatic and qualitative evaluation in a diverse range of tasks (Suzgun et al., 2023), including NMT (Freitag et al., 2022a) and code generation (Shi et al., 2022). MBR also generalizes other previously proposed decoding methods and explains their success (Bertsch et al., 2023).

The accuracy improvement from MBR comes at a heavy cost: the number of samples used can reach thousands (Freitag et al., 2023), and the number of calls to the utility function required is quadratic in the number of samples. Often, the utility function itself is a deep neural model, rendering MBR prohibitively expensive for many use cases.

In this work, we address the computational efficiency of MBR with an iterative pruning algorithm where low-performing hypotheses are removed while the number of samples used to estimate utilities grows. Hypotheses are pruned based on their estimated probability of being the true best hypothesis under the MBR objective, thus avoiding making expensive fine-grained utility estimates for hypotheses which are unlikely to be the final prediction.

In NMT experiments on three language pairs using chrF++ (Popović, 2015), and COMET (Rei et al., 2020) as MBR utility and evaluation metrics, we show that our method maintains the same level of accuracy as standard MBR while reducing the number of utility calls by a factor of at least 7 for chrF++ and 15 for COMET. Our algorithm can also use fewer samples to reach a prediction by terminating early, unlike standard MBR.

## 2 Minimum Bayes risk decoding

Conditional sequence generation problems such as neural machine translation (NMT) model the probability of the next token $y_t$ given a source sequence $x$ and prefix $y_{<t}$ with a neural network $p_\theta$. This

model can be used to assign probabilities to full sequences $p_\theta(y|x)$ via the chain rule.

At test time, a decision rule is employed to select a single "best" sequence. The most common decision rule is to return the highest probability sequence $y^{MAP} = \arg\max_y p_\theta(y|x)$. The exact solution is generally intractable, and beam search is used to find an approximation.

In contrast, minimum Bayes risk decoding (MBR) (Goel and Byrne, 2000) outputs:

$$y^{MBR} = \arg\max_y \mathbb{E}_{\bar{y}\sim p_\theta(\cdot|x)}[u(y,\bar{y})]$$
$$= \arg\max_y U(y, p_\theta(\cdot|x)), \qquad (1)$$

for some utility function $u$, a measure of similarity between sequences, and $U(y, \mathcal{Y}) = \mathbb{E}_{\hat{y}\sim\mathcal{Y}}[u(y,\hat{y})]$, where $\mathcal{Y}$ is either a probability distribution or an array of samples. We call $U$ the *expected utility function*. Eikema and Aziz (2020) propose a sampling method for neural language models where hypotheses $\mathcal{H}$ and pseudo-references $\mathcal{R}$ are generated with unbiased sampling, and $y^{MBR}$ is estimated as:

$$y^{MBR} \approx \arg\max_{y\in\mathcal{H}} U(y, \mathcal{R}). \qquad (2)$$

This method, which we refer to as "standard" MBR, requires $|\mathcal{H}|+|\mathcal{R}|$ samples (assuming $\mathcal{H} \neq \mathcal{R}$) and $|\mathcal{H}||\mathcal{R}|$ calls to $u$. The latter is the main computational bottleneck which we address in this work. Recent works on MBR focus on identifying accurate and efficient generation methods (Eikema and Aziz, 2021; Freitag et al., 2023; Yan et al., 2022), and pruning $\mathcal{H}$ to a smaller size with a faster method prior to running standard MBR (Eikema and Aziz, 2021; Fernandes et al., 2022).

Freitag et al. (2022a) and Eikema and Aziz (2021) show that MBR is more effective relative to MAP when the utility metric has high segment-level accuracy as measured by Freitag et al. (2021). So, we use COMET (Rei et al., 2020), one of the best available metrics for NMT, and chrF++ (Popović, 2015), as a simpler and faster yet reasonably good lexical metric. We heed the call of Freitag et al. (2022b) and do not use BLEU, which is obsoleted by newer metrics as both an evaluation and utility metric (Freitag et al., 2022a).

## 3 Confidence-based hypothesis pruning

Sampling-based MBR returns the highest-utility hypothesis from a set measured over pseudo-references sampled from the model. Speedups can be achieved if low-performing hypotheses are removed from consideration based on coarse utility estimates obtained from a subset of the pseudo-references. In other words, we can save time by not computing precise utility estimates for hypotheses which are unlikely to be chosen in the end.

We propose an iterative algorithm for MBR where the hypothesis set is gradually shrunk while the pseudo-reference list grows. The procedure is shown in Algorithm 1. We start with an initial hypothesis set $\mathcal{H}_1$, and at each time step $t$, a pruning function uses a pseudo-reference list $\mathcal{R}_t$ of size $r_t$ to select $\mathcal{H}_{t+1} \subseteq \mathcal{H}_t$. After the maximum time step is reached or when the current hypothesis set contains one element, terminate and return the highest utility hypothesis under all available pseudo-references. The size of $\mathcal{R}_t$ grows according to a pre-defined "schedule" $r_1, ..., r_T$.

---

**Algorithm 1** Pruning MBR

**Input:** Source sentence $x$.
**Constants:** sample size schedule $r = r_1, ..., r_T$, expected utility function $U$, model parameters $\theta$, pruning function prune, hypothesis generation function $\text{gen}(x, \theta)$.
**Output:** An MBR prediction.
1: $\mathcal{R}_0 \leftarrow [\,]$
2: $t \leftarrow 1$
3: $\mathcal{H}_1 \leftarrow \text{gen}(x, \theta)$
4: **while** $t \leq T$ and $|\mathcal{H}_t| > 1$ **do**
5: $\quad \mathcal{R}_t \leftarrow \mathcal{R}_{t-1}$
6: $\quad$ **while** $|\mathcal{R}_t| < r_t$ **do**
7: $\quad\quad$ Append $\hat{y} \sim p_\theta(\cdot|x)$ to $\mathcal{R}_t$
8: $\quad$ **end while**
9: $\quad \mathcal{H}_{t+1} \leftarrow \text{prune}(\mathcal{H}_t, \mathcal{R}_t)$
10: $\quad t \leftarrow t + 1$
11: **end while**
12: **return** $\arg\max_{y\in\mathcal{H}_t} U(y, \mathcal{R}_{t-1})$

---

The goal of the pruning function is to exclude as many hypotheses as possible to reduce the number of utility calls made in the future without excluding the true top-ranking MBR hypothesis $\arg\max_{y\in\mathcal{H}_t} U(y, p_\theta(\cdot|x))$, the true "winner".

We propose to prune hypotheses in $\mathcal{H}_t$ with low probability of being the true winner and to estimate this probability using nonparametric bootstrap resampling (Efron, 1979; Koehn, 2004); given an initial collection of i.i.d. samples $\mathcal{S}$ from an unknown distribution $\mathcal{X}$, our beliefs about the true value of any statistic $T(\mathcal{X})$ are represented by the distribu-

tion $p(T(\hat{\mathcal{S}}))$, where $\hat{\mathcal{S}} \sim \text{boot}(\mathcal{S})$, and $\text{boot}(\mathcal{S})$ returns a with-replacement size-$|\mathcal{S}|$ resample of $\mathcal{S}$.

In our case, we want to estimate the probability that $y$ is the true winner in $\mathcal{H}_t$:

$$p\big( \bigwedge_{\bar{y} \in \mathcal{H}} U(y, p_\theta(\cdot|x)) \geq U(\bar{y}, p_\theta(\cdot|x)) \big), \quad (3)$$

which we estimate as the chance that $y$ wins in a bootstrap resample. Let $\hat{\mathcal{R}}_t \sim \text{boot}(\mathcal{R}_t)$. Then the bootstrap estimator is:

$$\mathbb{E}_{\hat{\mathcal{R}}_t \sim \text{boot}(\mathcal{R}_t)} \big[ 1 \bigwedge_{\bar{y} \in \mathcal{H}_t} (U(y, \hat{\mathcal{R}}_t) \geq U(\bar{y}, \hat{\mathcal{R}}_t)] . \quad (4)$$

This estimator can be high variance because the probability $y$ winning in a bootstrap sample is very small when $\mathcal{H}_t$ is large, so instead we use the probability that $y$ outranks a *particular* $\bar{\bar{y}} \in \mathcal{H}_t$:

$$\mathbb{E}_{\hat{\mathcal{R}}_t \sim \text{boot}(\mathcal{R}_t)} \big[ 1(U(y, \hat{\mathcal{R}}_t) \geq U(\bar{\bar{y}}, \hat{\mathcal{R}}_t)) \big] . \quad (5)$$

This statistic is invariant to the size of $\mathcal{H}_t$ because it only considers utility estimates of $y$ and $\bar{\bar{y}}$. It is an upper bound of Equation 4 because the probability of $y$ winning against all $\bar{y} \in \mathcal{H}_t$ cannot be higher than the probability of winning against a particular $\bar{y} \in \mathcal{H}_t$. $\bar{\bar{y}}$ can be any element in $\mathcal{H}_t$, but we set it to the winner under $\mathcal{R}_t$, i.e. $\bar{\bar{y}} = \arg\max_{\bar{y} \in \mathcal{H}_t} U(\bar{y}, \mathcal{R}_t)$, to achieve a tighter upper bound.

We propose to prune $y$ if its estimated probability of beating $\bar{\bar{y}}$ is less than $1 - \alpha$, where $\alpha$ is a confidence threshold $0 \leq \alpha \leq 1$. In summary, this procedure prunes some but not necessarily all $y \in \mathcal{H}_t$ which are estimated to have less than $1 - \alpha$ chance of being the true winner. Algorithm 2 shows the procedure in detail.

Note that bootstrapped utility estimates do not require additional utility function calls because they reuse utility values already computed over $\mathcal{H}_t, \mathcal{R}_t$. Also note that the bootstrap estimator is always biased because $\mathcal{R}_t$ is never equal to $p_\theta(\cdot|x)$, and the bias is worse when $\mathcal{R}_t$ is small. Nonetheless, we show empirically that bootstrapping is effective in our pruning algorithm for modest sizes of $\mathcal{R}_t$.

Another benefit of our pruning method compared to standard MBR is that it can terminate early if $\mathcal{H}_t$ has only one remaining hypothesis, reducing the total number of pseudo-references needed.

As a baseline pruning function for comparison, we rank each $y \in \mathcal{H}_t$ by $U(y, \mathcal{R}_t)$ and prune the

---

**Algorithm 2** Confidence-based pruning function

**Input:** Hypothesis set $\mathcal{H}$, pseudo-reference list $\mathcal{R}$.
**Constants:** Expected utility function $U$, confidence threshold $\alpha$, number of bootstrap samples $n$.
**Output:** A subset of $\mathcal{H}$.

1: $\mathcal{H}_{new} \leftarrow \{\}$
2: $\bar{\bar{y}} \leftarrow \arg\max_{\bar{y} \in \mathcal{H}} U(\bar{y}, \mathcal{R})$
3: $\mathcal{H}_1 \leftarrow \text{gen}(x, \theta)$
4: **for** $i \leftarrow 1$ to $n$ **do**
5: $\quad \hat{\mathcal{R}}^i \leftarrow \text{boot}(\mathcal{R})$
6: **end for**
7: **for** $y \in \mathcal{H}$ **do**
8: $\quad w \leftarrow \frac{1}{n} \sum_i^n 1(U(y, \hat{\mathcal{R}}^i) \geq U(\bar{\bar{y}}, \hat{\mathcal{R}}^i))$
9: $\quad$ **if** $w > 1 - \alpha$ **then**
10: $\quad\quad \mathcal{H}_{new} \leftarrow \mathcal{H}_{new} \cup \{y\}$
11: $\quad$ **end if**
12: **end for**
13: **return** $\mathcal{H}_{new}$

---

bottom-$\beta$ proportion. At $\beta = 0$, no pruning occurs and standard MBR decoding is recovered. We refer to this baseline as $\text{prune}_\beta$ and our confidence-based method as $\text{prune}_\alpha$.

## 4 Experiments

We perform our experiments on NMT models which we train on the German-English (de-en), English-Estonian (en-et), and Turkish-English (tr-en) news datasets from WMT18. We use the data preprocessing steps provided by the WMT18 organizers, except that we exclude Paracrawl from the de-en dataset following Eikema and Aziz (2021). The final training sets have 5.8 million, 1.9 million, and 207 thousand sentence pairs respectively. All models are transformers of the base model size from Vaswani et al. (2017) and are trained without label smoothing until convergence.

For all language pairs and validation/test datasets, we generate $\mathcal{H}_1$ with beam top-$k$ with $k = 256$[1]. We generate 1024 pseudo-references $\mathcal{R}^*$ with epsilon sampling (Hewitt et al., 2022) with $\epsilon = 0.02$, following Freitag et al. (2023). In order to run multiple random trials efficiently, we simulate sampling pseudo-references by sampling from $\mathcal{R}^*$ without replacement. The experiments in Sections 4.1 and 4.2 show averaged results from 10 trials. We use chrF++ and COMET as utility func-

---

[1]Different sequences may have the same detokenization result, so we have fewer than 256 hypotheses on average.

tions and always match the evaluation metric to the utility. chrF++ is computed using SacreBLEU (Post, 2018) with default settings. COMET is computed with the COMET-22 model from Rei et al. (2022). We use 500 bootstrap samples for $\text{prune}_\alpha$.

For the pseudo-reference sample size schedule $r_1, ..., r_T$, the choice of $r_1$ is pivotal in the speed-accuracy trade-off; $|\mathcal{H}_1||\mathcal{R}_1|$ is a lower bound on the number of utility calls needed, but the bootstrap estimate is more biased when the sample size is small. In a preliminary experiment on the validation set, we measure the "false pruning rate", the rate that the estimated true winner, $\arg\max_{\bar{y}\in\mathcal{H}} U(\bar{y}, \mathcal{R}^*)$ is pruned under different choices of $\alpha$ and $|\mathcal{R}|$. Based on results shown in Figure 1, we set $r_1$ to 8 for COMET and 16 for chrF++ for all experiments. $r_2, ..., r_T$ are set by doubling at each time step until reaching 256.

More experimental details and figures for language pairs not shown here are in the Appendix. Our code is publicly available[2].

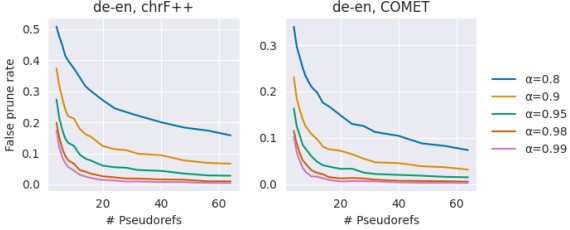

Figure 1: False pruning rates for different choices of $\alpha$ and $|\mathcal{R}|$ measured on the validation set.

## 4.1 Speed-accuracy trade-off

$\text{prune}_\alpha$ and $\text{prune}_\beta$ allow control over the speed-accuracy trade-off with a single parameter. We observe this trade-off over the validation set by comparing the number of utility function calls made against various measures of accuracy. High evaluation score is underlying goal, but we find that the score changes very little across settings, so we also evaluate pruning quality in terms of how well final predictions match those under standard MBR with $\mathcal{R}^*$. We use *exact accuracy*, whether the prediction $y$ equals $\arg\max_{\bar{y}\in\mathcal{H}} U(\bar{y}, \mathcal{R}^*)$, and *reciprocal rank* (RR), equal to $(\sum_{\bar{y}\in\mathcal{H}} 1(U(\bar{y}, \mathcal{R}^*) \geq U(y, \mathcal{R}^*)))^{-1}$ as a soft accuracy measure adapted from the mean reciprocal rank used in search (Craswell, 2009). Figure 2 shows that $\text{prune}_\alpha$ generally outperforms $\text{prune}_\beta$ on all metrics.

[2]https://github.com/juliusc/pruning_mbr

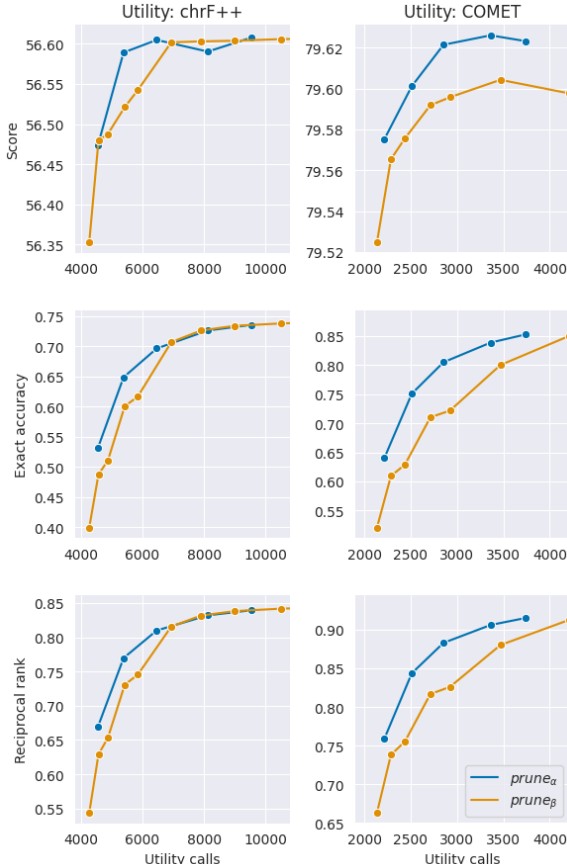

Figure 2: Speed-accuracy trade-off curves of pruning functions for $\alpha \in 0.8, 0.9, 0.95, 0.98, 0.99$ and $\beta \in \{0.05, ..., 0.95\}$ on the de-en validation set. The x-axes are truncated for better visual comparison.

## 4.2 Test results

We evaluate our methods on the test set with $\alpha = 0.99$ and $\alpha = 0.9$ and compare them to standard MBR on the accuracy metrics described in Section 4.1 as well as the number of utility calls and pseudo-references used. Table 1 shows that across all language pairs and metrics, our method achieves similar evaluation scores as standard MBR while using much less computation, with the most dramatic case being en-et and COMET with $\alpha = 0.9$ which uses 3.5% as many utility calls and 32% as many pseudo-references as the baseline.

When comparing $\alpha = 0.99$ with $\alpha = 0.9$, we see that while exact accuracy and RR differ, the evaluation score differs very little if at all, suggesting that high-ranking hypotheses are often equally good as one another, and finely discriminating between them has diminishing returns.

| | Metric: chrF++ | | | | | | | | |
|---|---|---|---|---|---|---|---|---|---|
| | de-en | | | en-et | | | tr-en | | |
| | $\beta = 0$ | $\alpha = 0.99$ | $\alpha = 0.9$ | $\beta = 0$ | $\alpha = 0.99$ | $\alpha = 0.9$ | $\beta = 0$ | $\alpha = 0.99$ | $\alpha = 0.9$ |
| Score | 62.18 | 62.19 | 62.11 | 46.47 | 46.48 | 46.44 | 42.63 | 42.63 | 42.64 |
| Accuracy | 0.749 | 0.736 | 0.648 | 0.741 | 0.728 | 0.639 | 0.761 | 0.751 | 0.660 |
| RR | 0.850 | 0.840 | 0.769 | 0.843 | 0.833 | 0.762 | 0.857 | 0.850 | 0.779 |
| # Pseudo-refs | 256.0 | 234.9 | 185.2 | 256.0 | 235.3 | 188.5 | 256.0 | 238.7 | 185.7 |
| # Utility calls | 65082 | 9542 | 5402 | 63898 | 9644 | 5386 | 65356 | 8761 | 5275 |
| | Metric: COMET | | | | | | | | |
| | de-en | | | en-et | | | tr-en | | |
| | $\beta = 0$ | $\alpha = 0.99$ | $\alpha = 0.9$ | $\beta = 0$ | $\alpha = 0.99$ | $\alpha = 0.9$ | $\beta = 0$ | $\alpha = 0.99$ | $\alpha = 0.9$ |
| Score | 78.37 | 78.37 | 78.41 | 82.75 | 82.74 | 82.74 | 73.70 | 73.68 | 73.65 |
| Accuracy | 0.872 | 0.844 | 0.742 | 0.924 | 0.898 | 0.830 | 0.897 | 0.878 | 0.791 |
| RR | 0.930 | 0.911 | 0.838 | 0.959 | 0.943 | 0.898 | 0.944 | 0.932 | 0.873 |
| # Pseudo-refs | 256.0 | 200.3 | 120.4 | 256.0 | 151.4 | 82.6 | 256.0 | 177.8 | 100.8 |
| # Utility calls | 65082 | 3831 | 2533 | 63898 | 2813 | 2250 | 65356 | 3244 | 2394 |

Table 1: Statistics for MBR decoding on the test set for all language pair and metric settings. $\beta = 0$ indicates standard MBR. All values are averaged across 10 random trials.

## 4.3 Human evaluation

We confirm that our method is indistinguishable from standard MBR in human evaluation. On the de-en test set, for each instance, we sample 256 pseudo-references without replacement from $\mathcal{R}^*$ and use this pool to decode with both standard MBR and $\mathrm{prune}_\alpha, \alpha = 0.99$. 85% of the predictions are the same, and for the rest, we asked bilingual readers of German and English to state which prediction they preferred. We obtained 125 ratings. The standard MBR prediction won in 48 cases, lost in 42, and tied in 35. This fails the significance test of Koehn (2004), so we conclude that $\mathrm{prune}_\alpha$ with $\alpha = 0.99$ is not significantly different from standard MBR on de-en.

## 4.4 Run times

To measure realistic run times, we implement a practical version of our algorithm and compare our method against standard MBR decoding and beam search. We run our algorithm with COMET as the utility function and $\alpha = 0.99$. With COMET, sentence embeddings can be cached, which greatly speeds up utility function calls. Some of the theoretical gains seen in Section 4.2 are diminished in practice due to our iterative algorithm dividing computations over smaller batches. Our best implementation samples all 256 pseudo-references at once but generates pseudo-reference sentence embeddings as needed. We run each algorithm on a set of 100 randomly sampled test instances. Further

implementation details are given in Appendix A.2. Table 2 provides a detailed breakdown of run times. As expected, our method achieves the greatest time savings on utility computation. However, it is still substantially slower than beam search.

| | Run time (seconds) | | |
|---|---|---|---|
| | Prune MBR | Std. MBR | Beam |
| Generate $\mathcal{H}$ | 36.56 | 36.56 | 26.79 |
| Generate $\mathcal{R}$ | 59.92 | 59.92 | |
| Embed sentences | 100.50 | 108.33 | |
| Compute utilities | 12.96 | 193.01 | |
| Bootstrap pruning | 0.63 | | |
| Total | 210.57 | 397.82 | 26.79 |

Table 2: Summary of run times for decoding methods on 100 sentences from the de-en test set for pruning MBR ($\alpha = 0.99$), standard MBR, and beam search with beam size 10, averaged over 3 trials.

## 5 Conclusion

We propose an iterative pruning algorithm for MBR along with a pruning criterion based on confidence estimates derived from bootstrap resampling. In experiments across diverse language pairs and metrics, we show that our method consistently outperforms our proposed baseline and achieves significant computational savings over standard sampling-based MBR without sacrificing accuracy. Our method is a drop-in substitute for standard MBR that requires no knowledge about the model $p_\theta$, how $\mathcal{H}_1$ is generated, or the utility function.

## Limitations

Even with our pruning algorithm, MBR is many times more costly to run than beam search. More

An important hyperparameter in our method is the sample size schedule. We show why it is important to carefully choose the size of the first sample, but not how the remaining schedule should be set, opting to simply double the size at each step. We leave this issue to future work.

Methods such as MBR and reranking that directly optimize a metric may exploit noise in the metric to improve the score without actually improving quality (Fernandes et al., 2022). In these settings, automatic evaluation is less trustworthy and should ideally be combined with human evaluation. However, human evaluation is difficult and expensive to obtain.

## Acknowledgements

Julius Cheng is supported by a scholarship from Huawei. The authors would like to thank the bilingual readers who helped with the human evaluation and the anonymous reviewers for their helpful suggestions.

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

## A  Additional experimental details

### A.1  All experiments

Preliminary experiments showed no significant difference between 500 and 1000 bootstrap samples when running $\text{prune}_\alpha$, so we use 500 for all experiments.

For efficiency, we use average sentence-level chrF++ instead of corpus-level chrF++ for corpus-level evaluations. This allows us to pre-compute the sentence-level chrF++ for each hypothesis and

obtain the corpus-level score of a set of predictions by simple averaging.

All experiments are implemented on top of our fork of Fairseq (Ott et al., 2019).

### A.2  Run times

This section contains additional details for the experiment in Section 4.4.

For both the standard and pruning MBR algorithms, we deduplicate and cache computations whenever possible. For each unique pseudo-reference, its sentence embedding and utility scores against each $y \in \mathcal{H}_t$ are only computed once.

For simplicity, all decoding methods are run on one sequence at a time. Batching across sequences would likely affect the relative performance characteristics of each method.

All experiments are conducted on the same machine with one Nvidia Quadro RTX 8000 GPU.

## B  False pruning rates for en-et, tr-en

Figure 3 shows the false pruning rates for en-et and tr-en.

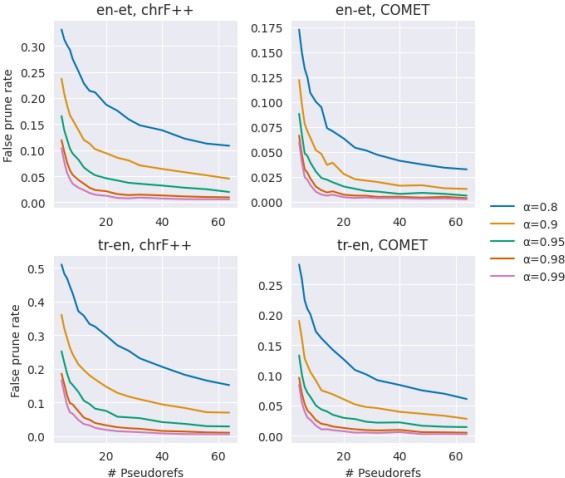

Figure 3: False pruning rates for different choices of $\alpha$ and $|\mathcal{R}|$ measured on the validation set.

## C  Hypotheses remaining per time step

Figure 4 shows the distribution of the number of remaining hypotheses after each time step when running our method on the de-en validation set, following the experimental setup of Section 4.1 where $|\mathcal{H}_1| = 256$. This is provided to further illustrate the pruning process.

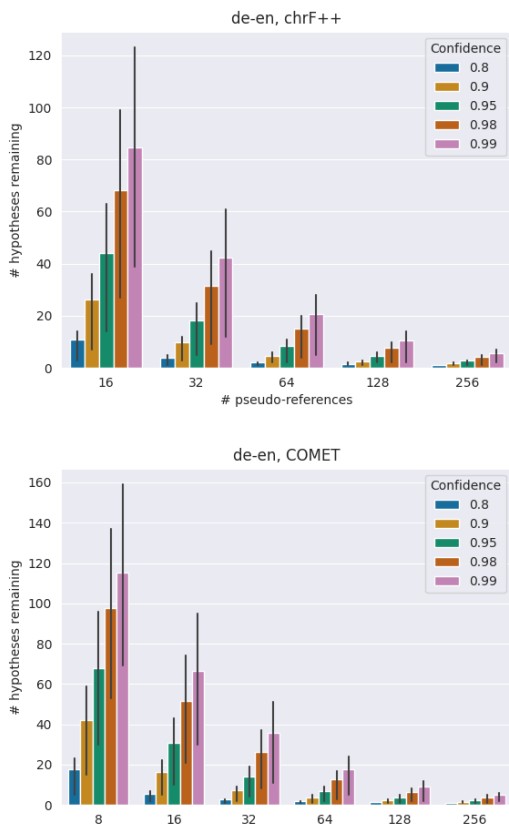

Figure 4: Number of remaining hypotheses after each time step while running the pruning MBR algorithm for various choices of $\alpha$ on the de-en validation set. The x-axis is the number of pseudo-references at a time step, and the y-axis is the number of hypotheses remaining after pruning. Colored bars show the mean, and error bars show the interquartile range.

## D  Speed-accuracy trade-off for en-et, tr-en

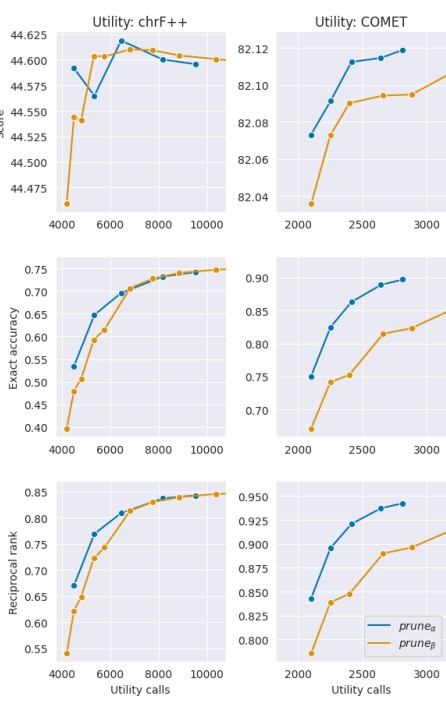

Figure 5: Speed-accuracy trade-off curves of pruning functions for $\alpha \in 0.8, 0.9, 0.95, 0.98, 0.99$ and $\beta \in \{0.05, ..., 0.95\}$ on the en-et validation set.

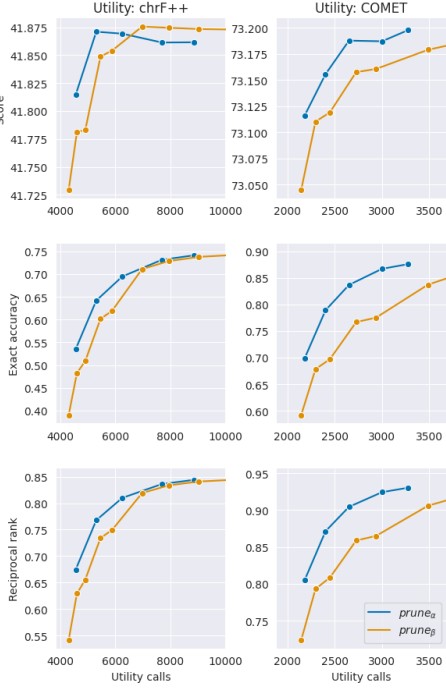

Figure 6: Speed-accuracy trade-off curves of pruning functions for $\alpha \in 0.8, 0.9, 0.95, 0.98, 0.99$ and $\beta \in \{0.05, ..., 0.95\}$ on the tr-en validation set.