# OpenReview forum: "Faster Minimum Bayes Risk Decoding with Confidence-based Pruning"
_EMNLP/2023/Conference — EMNLP 2023 Main_

### Official Review · Reviewer_urSQ · 2023-07-31

**Soundness:** 5

**Excitement:**

4: Strong: This paper deepens the understanding of some phenomenon or lowers the barriers to an existing research direction.

**Paper Topic And Main Contributions:**

This paper proposes a new iterative algorithm for performing minimum Bayes risk decoding, and does experiments on machine translation.

The basic idea behind the algorithm is to use bootstrapping to estimate the probability that a particular hypothesis will end up being the final winner.

**Reasons To Accept:**

Overall the work is:
1. tackling an important problem of improving the efficiency of MBR. MBR is a very effective and flexible method to improve accuracy of systems but its time efficiency is a major downside.
2. the formulation of this is elegant and statistically well grounded.
3. experimentation is thorough and covers the important bases given that the paper is short.

I think this is a very nice paper, and could be considered for an outstanding paper award.

**Reasons To Reject:**

I do not have any reason to reject this paper. It is well written and solid, especially for a short paper. I will likely advocate strongly for its acceptance (unless I'm missing something).

The only thing that I found somewhat dissatisfying was that it would have been nice to have a wall-clock time comparison, but there are so many factors that can go into that (how well the decoding algorithms are optimized, and how well the utility function calculations are implemented), so I think the paper is justified without these.

There was no mention of whether the code would be released. Hopefully the authors will release the code to improve reproducibility. However, the methodology is straightforward enough that I think the methods could be reproduced without.

**Reproducibility:**

3: Could reproduce the results with some difficulty. The settings of parameters are underspecified or subjectively determined; the training/evaluation data are not widely available.

**Reviewer Confidence:**

4: Quite sure. I tried to check the important points carefully. It's unlikely, though conceivable, that I missed something that should affect my ratings.

**Typos Grammar Style And Presentation Improvements:**

I have one major suggestion for a presentation improvement. I was actually quite confused at first by Equations (3) and (4) due to their unusual notation. Normally (I think) the "for all" sign normally comes before the term it is applied to, so I would rearrange the equation for clarity.

---

> ### Author Rebuttal · Authors · 2023-08-29
>
> Thank you for the review and for the encouraging words!
>
> > The only thing that I found somewhat dissatisfying was that it would have been nice to have a wall-clock time comparison [...]
>
> The current set of experiments were done in a simulated mode where hypothesis, pseudo-references, and utilities are all generated in advance (in a way that doesn’t affect the validity of our work) but we agree that this would be important to include, so we implemented a practical version of our algorithm and logged timings. Due to time constraints, we executed our yet-unoptimized implementation on 60 random examples from the de-en test set using COMET as the utility metric.
>
> |                | Standard | Pruning | Beam search (k=5) |
> | -------------- | -------- | ------- | ----------------- |
> | Generate $\mathcal{H}$        | 20.7     | 20.4    | 14.5              |
> | Generate $\mathcal{R}$        | 33.1     | 74.7    |                   |
> | Get sentence embeddings | 61.7     | 48.3    |                   |
> | Compute utilities      | 116.5    | 5.0     |                   |
> | Compute bootstrap      |          | 0.3     |                   |
> | Total               | 232.1    | 148.7   | 14.5              |
>
> As expected, the most dramatic gains are in the “compute utilities” row which is linearly related to the number of utility calls shown in Table 1. There are some optimizations that we will make: 1) we have not deduplicated sequences in the utility computation, and 2) the source-side encoding is recomputed for each iteration in the iterative pruning algorithm, which is why generating $\mathcal{R}$ is slower in pruning than in standard MBR. If accepted, we will make these changes and run it on the full test set for the camera-ready.
>
> > There was no mention of whether the code would be released. Hopefully the authors will release the code to improve reproducibility.
>
> This was an oversight on our part. We will release the code if accepted.
>
> >I have one major suggestion for a presentation improvement. I was actually quite confused at first by Equations (3) and (4) due to their unusual notation [...]
>
> Reviewer GsRs04 also found this confusing, and in response to both of your suggestions, we would change Equation 3 to:
>
> $p\bigl(\bigwedge\_{\bar{y}\in\mathcal{H}} U (y, p_\theta(·|x)) ≥ U (\bar{y}, p_\theta(·|x))\bigr)$
>
> and Equation 4 to:
>
> $\frac{1}{n}\sum^{n}_{i=1}{1 \bigwedge\_{\bar{y}\in\mathcal{H}_t}{U(y,\hat{\mathcal{R}^i_t})\geq U(\bar{y},\hat{\mathcal{R}^i_t})}}$
>
> and Equation 5 to:
>
> $\frac{1}{n}\sum^{n}\_{i=1}{1({U(y,\hat{\mathcal{R}^i_t})\geq U(\tilde{y},\hat{\mathcal{R}^i_t})})}$ where $\tilde{y}=\arg\max_{\bar{y}\in\mathcal{H}_t}U(\bar{y}, \mathcal{R}_t)$

---

### Official Review · Reviewer_JEUK · 2023-08-01

**Soundness:** 5

**Excitement:**

4: Strong: This paper deepens the understanding of some phenomenon or lowers the barriers to an existing research direction.

**Paper Topic And Main Contributions:**

This paper proposes an iterative pruning algorithm to reduce the number of hypotheses considered in minimum Bayes risk decoding (MBR) while gradually growing the number of candidate samples for utility estimation, for the task of neural machine translation.

The authors estimate the probability of a hypothesis being the winner among all hypotheses in question with nonparametric bootstrap resampling. In Layman's terms, at a certain iteration of the algorithm, a hypothesis has a chance to be pruned when it does not beat a certain other hypothesis often enough, when comparing with different sets of references.

The proposed algorithm significantly reduces the number of utility calls, while maintaining similar scores under the same utilitiy function. This is especially useful if each utility call gets expensive, e.g. querying a neural model to compare two translations.

**Questions For The Authors:**

One thing that is not crystal clear to me is why the pseudo-reference list grows. Is it because that the initial list contains not so good translations (the ones that will get pruned)? Is H1 and R1 the same? Does sampling more references and adding them to R improve the quality of the reference set? If so, why not also prune away those sentences that get pruned away?

Another question is about Figure 2, top-right sub-figure. Why do the scores go down in the end? Do I understand correctly that the final hypothesis has to come from H1 because no new hypothesis is ever added in later iterations? If so, does that mean at utility call 3500 for prune_alpha, the best hypothesis so far still gets somehow thrown away with more utility calls? How often is the true reference included in your pseudo reference set? I think one more curve in the same figure which is the number of remaining hypotheses (or maybe also add the current reference set size) would help better understand the process.

**Reasons To Accept:**

The paper is very well-written. The problem -> motivation -> existing solutions -> proposed solution -> explaining the solution -> making compromises for practical uses -> showing automatic results -> extending to human evaluations logic line is super clear and easy-to-follow.

The experiments show convincing results which clearly support the claims of the authors, that the proposed pruning method significantly reduces utility calls without hurting the translation accuracy.

**Reasons To Reject:**

Can't think of any. I like this paper.

**Reproducibility:**

4: Could mostly reproduce the results, but there may be some variation because of sample variance or minor variations in their interpretation of the protocol or method.

**Reviewer Confidence:**

4: Quite sure. I tried to check the important points carefully. It's unlikely, though conceivable, that I missed something that should affect my ratings.

**Typos Grammar Style And Presentation Improvements:**

L085 and L090, maybe not use italic for MAP and MBR in the superscript.

Figure 2, maybe legends in each subplot? maybe add current hypotheses set size curves.

Table 1, I would prefer if three vertical bars are added to separate the test sets, I was confused to read beta, alpha, alpha and again beta the first time I saw the table. Also, why not include a row with the actual run times? I would expect linear reductions in runtimes with reductions in number of utility calls, but would be nice to see it. For that matter, maybe also add a column for a typical beam search setup for a more complete comparison?

---

> ### Author Rebuttal · Authors · 2023-08-29
>
> Thank you for your review, and we're glad to hear that you liked our paper! We'll respond point-by-point:
>
> > One thing that is not crystal clear to me is why the pseudo-reference list grows. Is it because the initial list contains not so good translations (the ones that will get pruned)? Is H1 and R1 the same? Does sampling more references and adding them to R improve the quality of the reference set? If so, why not also prune away those sentences that get pruned away?
>
> The literature is mixed on whether $\mathcal{H}$ and $\mathcal{R}$ should be the same, so for maximum generality, we don’t assume so.
>
> The larger $\mathcal{R}$ is, the higher the quality of approximation of the utility of each hypothesis, and it has been observed in practice that better approximations mean better test evaluation scores (Freitag et. al., 2022). So, our algorithm, like standard MBR, seeks fine-grained utility estimates for hypotheses, except that we prune away hypotheses which are statistically unlikely to be the final winner. We hope this answers your questions, and we'll incorporate these clarifications into the text.
>
> > Another question is about Figure 2, top-right sub-figure. Why do the scores go down in the end? Do I understand correctly that the final hypothesis has to come from H1 because no new hypothesis is ever added in later iterations? If so, does that mean at utility call 3500 for prune_alpha, the best hypothesis so far still gets somehow thrown away with more utility calls?
>
> You are correct that no new hypotheses are added in later iterations, and the reason for this downward curve is that utility scores and evaluation scores are highly but not perfectly correlated.  Some of the equivalent figures in the appendix for en-et and tr-en show similar phenomena. The dip mostly occurs on the right side of the graph, and we believe that this is because as the evaluation score converges, the randomness in the evaluation score sometimes exceeds the expected improvement for more utility calls. We will add more points on these graph as well as confidence bands on these graphs to illustrate this point if accepted.
>
> > How often is the true reference included in your pseudo reference set?
>
> In Freitag et. al., 2022, the true reference is shown to have very low probability under their model, so we assume that the true reference is almost never in our pseudo-reference sets for sentences over a certain length.
>
> > I think one more curve in the same figure which is the number of remaining hypotheses (or maybe also add the current reference set size) would help better understand the process.
>
> We agree that such a figure that shows the average number of hypotheses remaining per pseudo-reference list size would provide insight and intuition into the process, and we will add this in the main text for de-en and in the appendix for en-et and tr-en.
>
> Lastly, thank you for the suggestions for style and presentation, which we will incorporate.
>
> Markus Freitag, David Grangier, Qijun Tan, and Bowen Liang. 2021a. High quality rather than high model probability: Minimum bayes risk decoding with neural metrics. arXiv preprint arXiv:2111.09388.

---

### Official Review · Reviewer_GsRs · 2023-08-04

**Typos Grammar Style And Presentation Improvements:** NA
**Soundness:** 4

**Excitement:**

4: Strong: This paper deepens the understanding of some phenomenon or lowers the barriers to an existing research direction.

**Missing References:**

Here's another paper discussing the MBR in MT:
DC-MBR: Distributional Cooling for Minimum Bayesian Risk Decoding
https://arxiv.org/pdf/2212.04205.pdf

**Paper Topic And Main Contributions:**

This paper addresses one of the important problems for Minimum Bayesian Risk decoding, the efficiency problem.
They propose a novel bootstrapping-based method to dynamically truncate the hypothesis set.
Experiments show that their methods achieve similar results with standard MBR and require much less computational cost.

**Questions For The Authors:**

1. About Equation 5: How is this a lower bound for Equation 4? If I understand your notation correctly, equation 4 checks the number of resamples that y has a higher value of utility than any of y in the hypothesis set. Equation 5 checks the number of resamples that y has a higher value of utility than the highest utility -> y has the highest utility. Two equations look the same to me. I don't really understand this notation.
2. Descriptions about your methods and baseline methods are too short to understand. I would recommend an pseudo-code at least in your appendix to improve the readability.
3. how do you choose your hyper-parameter? Do you directly tune on the test set? If not, please clarify.

**Reasons To Accept:**

The problem authors addressed in this paper is crucial for real-world applications with MBR decoding.
The idea is inspiring and convincing, which simultaneously shrinks the hypothesis space and grows the reference space. The bootstrapping resample method fits well to the application and does not introduce extra computations.
The experiments show that their methods achieve significant speedup and maintain almost the same decoding performance.

**Reasons To Reject:**

I have some minor concerns about your methods and experiments. Please see the questions below.
If my questions and missing references are well-addressed, I am willing to give a higher score of soundness.

**Reproducibility:**

3: Could reproduce the results with some difficulty. The settings of parameters are underspecified or subjectively determined; the training/evaluation data are not widely available.

**Reviewer Confidence:**

5: Positive that my evaluation is correct. I read the paper very carefully and I am very familiar with related work.

---

> ### Author Rebuttal · Authors · 2023-08-29
>
> Thank you for the review! To respond:
>
> 1\. Equations 4 and 5 do indeed look the same, but in the text immediately after Equation 5, we define $\bar{y}$ as the highest utility hypothesis under the current pseudo-reference set, as you describe. We see that using $\bar{y}$ in both equations but defining them differently in the text can be misleading, and we would modify Equation 4 to the following:
>
> $\frac{1}{n}\sum^{n}_{i=1}{1 \bigwedge\_{\bar{y}\in\mathcal{H}_t}{U(y,\hat{\mathcal{R}^i_t})\geq U(\bar{y},\hat{\mathcal{R}^i_t})}}$
>
> and Equation 5 to:
>
> $\frac{1}{n}\sum^{n}\_{i=1}{1({U(y,\hat{\mathcal{R}^i_t})\geq U(\tilde{y},\hat{\mathcal{R}^i_t})})}$ where $\tilde{y}=\arg\max_{\bar{y}\in\mathcal{H}_t}U(\bar{y}, \mathcal{R}_t)$
>
> This makes explicit that Equation 4 compares $y$ against all $\bar{y}\in\mathcal{H}_t$ while Equation 5 only compares $y$ against a particular hypothesis.
>
> In response to why Equation 5 is an upper bound on Equation 4, in the former, y has to “win” against all other hypotheses in a bootstrap trial for that trial to contribute probability mass, and in the latter, y only has to win against a single other hypothesis. The probability of beating all hypotheses in a set cannot be higher than the probability of beating a single hypothesis in that same set. We will include this intuitive justification in the camera ready if accepted.
>
> 2\. We agree that pseudocode would greatly help readability, and if accepted, we would include two pseudocode boxes, one for iterative pruning MBR in general, and one for our proposed pruning function. Here is a preview. For iterative pruning, let $x$ be the source sentence, $p$ be the language model, $\mathcal{H}$ the starting hypothesis, $r=r_1,...,r_t$ the sample size schedule, and $\text{prune}$ a pruning function:
>
> $\text{PruningMBR}(x, p, \mathcal{H}_1, r, \text{prune}):$
>
> $\mathcal{R}_0 \leftarrow \\{\\}$
>
> for $t$ from 1 to $|r|$:
>
> &emsp; $\mathcal{R}_t \leftarrow \mathcal{R}\_{t-1}$
>
> &emsp; while $|\mathcal{R}_t| < r_t$:
>
> &emsp;&emsp; append $y\sim p(\cdot|x)$ to $\mathcal{R}_t$
>
> &emsp; $\mathcal{H}\_{t+1} \leftarrow \text{prune}(\mathcal{H}_t,\mathcal{R}_t)$
>
> &emsp; if $|\mathcal{H}|=1$:
>
> &emsp; &emsp; break
>
> return $\arg\max\_{y\in\mathcal{H}\_{t+1}}U(y, \mathcal{R}_t)$
>
> Our proposed function, given utility functions $u$ and $U$ as defined in the paper, number of bootstraps $n$, and confidence $\alpha$, is defined as:
>
> $\text{prune}_\alpha(\mathcal{H}, \mathcal{R})$:
>
> $ \tilde{\mathcal{H}} \leftarrow \\{\\}$
>
> $\tilde{y} \leftarrow \arg\max\_{y\in\mathcal{H}}U(y, \mathcal{R})$
>
> for $i$ from 1 to $n$:
>
> &emsp; $\hat{\mathcal{R}}^i \leftarrow \text{boot}(\mathcal{R})$
>
> for each $y\in\mathcal{H}$:
>
>  &emsp; if $\frac{1}{n}\sum^{n}\_{i=1}{1({U(y,\hat{\mathcal{R}^i})\geq U(\tilde{y},\hat{\mathcal{R}^i})})} \geq 1 - \alpha$:
>
> &emsp;&emsp; $\tilde{\mathcal{H}} \leftarrow \tilde{\mathcal{H}} \cup \\{y\\} $
>
> return $\tilde{\mathcal{H}}$
>
> 3\. There are two classes of hyperparameters. The first one contains those that are shared between the baseline and our method, and these are MT model training hyperparameters, methods for generating H and R, and utility function hyperparameters. The paper includes descriptions of non-standard choices for reproducibility.
>
> The second class contains those hyperparameters that are specific to our pruning method, namely the sample size schedule, the confidence threshold, and the number of bootstrap samples n. For the sample size schedule, we pick R_1 by balancing the false pruning rate seen in Figure 1 against how many utility calls are made on the validation set. The confidence threshold was not tuned as we show results for the full sweep. The number of bootstrap samples n is not critical; we tried n=500 and n=1000 in early experiments and the results were similar, and the runtime difference is small. We will make sure to include this information, especially the fact that the sample size schedule was tuned on the validation set.
>
> Re: the missing reference, this paper also pertains to MBR and we will include it in our overview of recent work on MBR.

---

### Meta-Review · Area_Chair_YbJZ · 2023-09-10

**Recommendation:** 5

**Metareview:**

This paper presents a bootstrapping method to dynamically truncate the hypothesis set in Minimum Bayesian Risk decoding.
The reviewers found this paper very exciting, appreciating the problem, the solution, the results, and the writing. There were very minor concerns raised and they were addressed.

---

### Decision · Program_Chairs · 2023-10-07

**Decision:**

Accept-Main

**Comment:**

This paper presents a bootstrapping method to dynamically truncate the hypothesis set in Minimum Bayesian Risk decoding.
The reviewers found this paper very exciting, appreciating the problem, the solution, the results, and the writing. There were very minor concerns raised and they were addressed.